# First demonstration of in-memory computing crossbar using multi-level Cell FeFET

Taha Soliman ⬥[1,7] ✉, Swetaki Chatterjee ⬥[2,3,7], Nellie Laleni[4], Franz Müller ⬥[4], Tobias Kirchner ⬥[1], Norbert Wehn[5], Thomas Kämpfe ⬥[4] ✉, Yogesh Singh Chauhan ⬥[3] & Hussam Amrouch ⬥[6] ✉

Advancements in AI led to the emergence of in-memory-computing architectures as a promising solution for the associated computing and memory challenges. This study introduces a novel in-memory-computing (IMC) crossbar macro utilizing a multi-level ferroelectric field-effect transistor (FeFET) cell for multi-bit multiply and accumulate (MAC) operations. The proposed 1FeFET-1R cell design stores multi-bit information while minimizing device variability effects on accuracy. Experimental validation was performed using 28 nm HKMG technology-based FeFET devices. Unlike traditional resistive memory-based analog computing, our approach leverages the electrical characteristics of stored data within the memory cell to derive MAC operation results encoded in activation time and accumulated current. Remarkably, our design achieves 96.6% accuracy for handwriting recognition and 91.5% accuracy for image classification without extra training. Furthermore, it demonstrates exceptional performance, achieving 885.4 TOPS/W—nearly double that of existing designs. This study represents the first successful implementation of an in-memory macro using a multi-state FeFET cell for complete MAC operations, preserving crossbar density without additional structural overhead.

Although the demand for data transmission is increasing globally, it is expected that the future will prioritize data-centric local intelligence at the edge node. This, in turn, will inevitably necessitate devices, including wearables, sensors, smartphones, and cars, to locally analyze data and make autonomous decisions.

Edge-AI devices have great potential to enable new applications with higher performance and support local embedded intelligence, real-time learning, and autonomy. This could propel the semiconductor industry's next growth phase. Energy-efficient local computing is crucial to enable smart connected Internet of Things (IoT) devices. Further, the rise of memory-intensive computational tasks has led to a significant increase in the amount of data that needs to be accessed compared to local computation inside the algorithmic logic units. This issue is commonly known as the von-Neumann bottleneck[1]. One of the most notable examples of such storage-bound tasks is related to AI, specifically when it comes to deep learning applications. In recent years, the research community has demonstrated the potential of various architectural improvements in computing systems, such as near-memory or in-memory-computing (IMC), to meet these energy, compute, and memory requirements[2].

Similarly, deep neural networks (DNNs) have gained popularity due to their remarkable performance, especially in applications such

[1]Robert Bosch GmbH, Renningen, Germany. [2]Semiconducture Test and Reliability, University of Stuttgart, Stuttgart, Germany. [3]Department of Electrical Engineering, Indian Institute of Technology Kanpur, Kanpur, India. [4]Fraunhofer IPMS, Dresden, Germany. [5]RPTU Kaiserslautern-Landau, Kaiserslautern, Germany. [6]Technical University of Munich; TUM School of Computation, Information and Technology; Chair of AI Processor Design; Munich Institute of Robotics and Machine Intelligence (MIRMI), Munich, Germany. [7]These authors contributed equally: Taha Soliman, Swetaki Chatterjee. ✉e-mail: taha.soliman@de.bosch.com; thomas.kaempfe@ipms.fraunhofer.de; amrouch@tum.de

as speech recognition and image processing. However, in order to design an efficient DNN, various metrics such as throughput, latency, and energy efficiency must be jointly optimized. Therefore, researchers have turned their attention to IMC architectures for deploying such networks. Such architectures perform MAC operations by utilizing the memory array without the need for data movement, resulting in improved system performance and large energy savings, thus, overcoming the fundamental bottleneck of von-Neumann architecture.

Advanced IMC architecture design relies on the usage of emerging memory cells, particularly non-volatile memory (NVM) cells. These NVMs are used to store weights in neural network inference architectures without needing a steady power supply and to perform multiply and accumulate (MAC) operations using their analog properties. Various emerging memories have been presented, such as resistive RAM (ReRAM), magnetic RAM, and FeFETs[3–5]. In this work, we focus on using FeFETs as memory cells due to their superior performance compared to other emerging memories[5], as it has been comprehensively summarized in the recent FeFET survey[6]. Our FeFET is co-integrated into a 28nm high-$\kappa$-metal-gate (HKMG) technology, resulting in a low footprint. The FeFET exhibits low read latency(~1ns), current source capability, and extremely high write power efficiency (<1 fJ) with a short write duration (~1μs), making it a highly suitable memory cell for IMC[5].

State-of-the-art FeFET based IMC architectures have been limited to binary logical operations, specifically logical AND and XNOR[7–11]. These operations are restricted to storing only two states within the FeFET memory cell. Even the use of multi-level cell (MLC) FeFETs will not offer any additional advantages in these architectures as the computations are limited to only binary operations. However, MLC FeFET have been utilized for other tasks such as matching in hyperdimensional computing[12]. We demonstrate for the first time a multi-bit MAC operation with variation-affected FeFET cells.

Contrary to analog computing based implementations of the MAC macro[13,14], we do not perform direct analog multiplication of the input and weight, which is highly prone to variations and requires a high degree of linearity in the stored states. Instead, we operate using the current-limited cell such that each cell that is activated has the same current contribution, which limits the impact of variation and improves operation accuracy. This is concisely what enabled us to overcome the variation and utilize MLC FeFET for the first time to demonstrate full MAC operations.

We designed and demonstrated a MAC circuit macro consisting of cells connected in the crossbar structure using the 1FeFET-1R configuration, which includes a single FeFET and a single resistor. In this macro, the FeFET cell acts as a memory for the entire weight value. We explore in this work three dimensions (time, stored $V_{th}$ state of the FeFET, and output current) to perform a complete MAC operation using the single FeFET per weight, detailed in Fig. 1. The input is encoded in applied voltage duration and magnitude, the multi-bit weight is stored in the FeFET, and the output is accumulated as the capacitor voltage that depends on the activation time and number of FeFETs activated as shown in Fig. 1.

Depending on the stored and input values, the multiplication operation result is encoded in the FeFET cell activation time. Additionally, the number of simultaneously activated FeFET cells is represented by the drain current over time which reflects the accumulation of the performed multiplications. Based on the voltage of the capacitor in the decoder block connected to an operating column of cells (as shown in Fig. 1h), the final MAC operation results can be determined and quantized to the required bit precision using a suitable analog-to-digital (ADC) converter. To prove the validity of our macro design, the stored weights, input, as well as output, are quantized to 2 bits. However, the presented macro and computational idea can be extended to higher precision as long as the memory cell can store the targeted

precision while maintaining the clear separation between the memory states.

We demonstrate in this paper the multiplication operation by a single FeFET cell and extend to complete MAC using the IMC macro of $32 \times 32$ FeFETs. This macro is tested against the LeNet network for MNIST handwritten digit database and several layers from VGG-19 for the CIFAR-10 dataset. Our presented work maintained the network accuracy at 96.6% with less than 2% accuracy loss. We leverage the properties of the FeFET device and demonstrate a novel architecture for multi-bit MAC operations enabling neural network inference within the memory to overcome the von-Neumann bottleneck. Thus, this work connects all the layers of the traditional computing stack from devices to architecture to system-level application.

## Our proposed architecture
### Single-cell multiply operation
The polarization of the ferroelectric layer can be modulated by the application of voltage pulses which in turn changes the $V_{th}$ of the underlying transistor and hence the conductance. The different $V_{th}$ states are achieved with respect to the value of the voltage pulse and the pulse duration. Figure 1b illustrates the MLC FeFET where it is programmed to four different states to store 2-bit information.

In this work, we use the 1FeFET-1R concept (to limit the drain current variability and operate in the saturation region) for a memory cell[8,12]. This concept is constructed by adding a drain resistance to the FeFET. The resistance is in the range of $1 M\Omega$ for a drain current of approximately 100 nA and can be implemented by a transistor biased at its gate or a physical resistance[12,14]. In this way, the current variability is reduced, as well as the power consumption of the memory unit cell. With the addition of the resistance, the current is limited to $0.1 \mu A$ as compared to $2 \mu A$ without the additional resistance. This reduces the cell power which is a product of the cell-current and the supply voltage.

We propose a novel methodology to use the MLC FeFET cell to perform the complete MAC operation. As shown in Fig. 2a, the 2-bit weight is encoded in the stored state inside the FeFET state such that the smallest value is represented by the highest $V_{th}$ state and the largest by the lowest $V_{th}$ state. On the other hand, the input value will be encoded as the time stamp at which the different gate voltages are applied according to Eq. (1)

$$2bit - V_g(Input, time) = \begin{cases} 0, & \text{if } Input.time < \epsilon_1, \\ V_{th1}, & \text{if } \epsilon_1 <= Input.time < \epsilon_2, \\ V_{th2}, & \text{if } \epsilon_2 <= Input.time < \epsilon_3, \\ V_{th3}, & \text{if } \epsilon_3 <= Input.time. \end{cases} \quad (1)$$

According to Eq. (1), the value of the gate voltage applied to the FeFET cell over time is dependent on the 2-bit input value. Consequently, the timestamp at which the FeFET saturates can encode the results of the multiplication of the stored state inside the FeFET as well as the input voltage. However, one of the main issues will be ensuring the commutative property for multiplication. In Eq. (1), the values $\epsilon_1, \epsilon_2, \epsilon_3$ are used to tune the ramping voltage speed to ensure the commutative property. As shown in Fig. 2, we can demonstrate a 2-bit/ 2-bit multiply operation performed by our memory cell in both simulations and measurements.

### Multiple-cell accumulation operation
To read out the multiple-cell accumulation operation, we utilized the charging of a capacitor due to the total accumulated drain currents in a column over a period of time. The drain current from multiple FeFETs are collected and directed to the accumulation capacitor. The saturation time of the FeFET encodes the multiplication output, and the number of FeFETs activated at a given time represents the

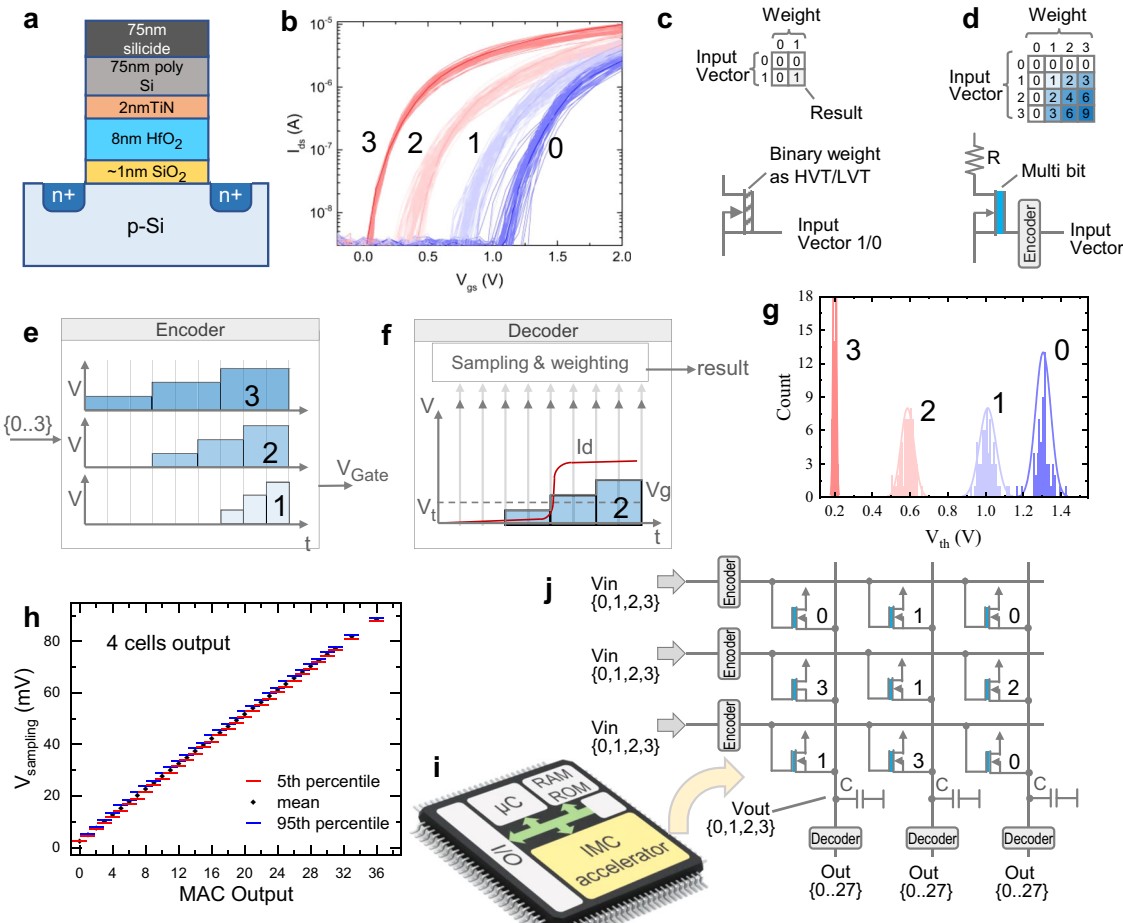

**Fig. 1 | Overview of the proposed IMC macro for MAC operations. a** The material stack of FeFETs. **b** The multi-bit FeFET can be programmed to different states to store the weight of the synapse. **c** Previous works[7, 8, 11] only considered binary AND or XNOR operations to compute a single-bit multiplication operation. **d** Our proposed 2-bit multiplication operation with input encoding and 2-bit storage is shown. The corresponding output activates at different instances of time. **e** An encoder provides the gate voltage depending on the input value which changes between three levels at different instances of time. **f** The multiplication output of the input and stored state in the cell depends on the time at which one cell is

activated which is accumulated and sampled using the decoder. **g** The $V_{th}$ distribution of the four states for the $I_{ds} - V_{gs}$ curves is shown. **h** Depending on the activation time and the number of cells activated at a given time, the voltage across the capacitor connected to a column of cells is accumulated which corresponds linearly with the MAC output and has a minimal impact of the underlying device variation. **i** IMC accelerators facilitate MAC operations for AI workloads where our proposed design can be utilized. **j** The corresponding MAC operation is performed in the crossbar, accumulating the output in the capacitor voltage.

accumulation. The capacitor voltage at the end of the period or the sampling time($t_s$) is given by

$$V_{sampling} = \frac{1}{C} \int_0^{t_s} \sum_{n=1}^{nFeFETs} I_{d,n}(t)\, dt \qquad (2)$$

where $C$ is the capacitance of the capacitor attached to the column, $I_{d,n}$ is the drain current of the $n^{th}$ FeFET, *nFeFETs* is the total number of FeFETs connected. At that point, the FeFETs are in the saturation region and limited by the resistance according to Eq. (3).

$$I_{d,n}(t) = \frac{V_{dd} - V_s(t)}{R_{out}} \cdot h(t - t_o) \qquad (3)$$

$V_s(t)$ is following the charging voltage function over a capacitance $C$ and time constant $\tau = R_{out} \cdot C$ where $R_{out}$ is the total resistance seen by the source of the FeFET including the current limiter resistance $R$. The time-constant $\tau$ is sufficiently high to not allow a steep fall in the current. $h(t - t_o)$ is the step-function at time $t_o$. It represents the time at which the FeFET is activated and it depends on the input applied and the stored state. The value of $t_o$ is such that it maintains the

commutative property of multiplication. Since the current follows a step-jump at $t_o$, the voltage across the capacitor follows a ramp after time $t_o$.

To quantize the MAC output, we employ StrongArm voltage input comparators[15] followed by latches and encoder for a complete 2-bit output. We explored the influence of the value of *nFeFETs* and the induced cell variation on the accumulation results and the resulting loss in inference accuracy.

## Results
### Experimental measurements
The FeFET test structures are fabricated in the GlobalFoundries 28 nm high-k/metal gate technology node, for which co-integration of FeFETs with CMOS devices has been demonstrated[16]. The FeFETs consist of a $SiO_2$ interfacial oxide layer, followed by an 8 to 10 nm thick, ferro-electric doped $HfO_2$ layer as illustrated in Fig. 1a and Fig. S1a in the Supplementary Materials. The gate is capped with a TiN metal cap and silicided poly silicon[17].

A 1FeFET-1R cell is constructed by externally connecting a 1 MΩ resistor to a FeFET with an area of 450 × 450 nm². This is necessary to limit the ON current and control variations. The 1FeFET-1R cell is

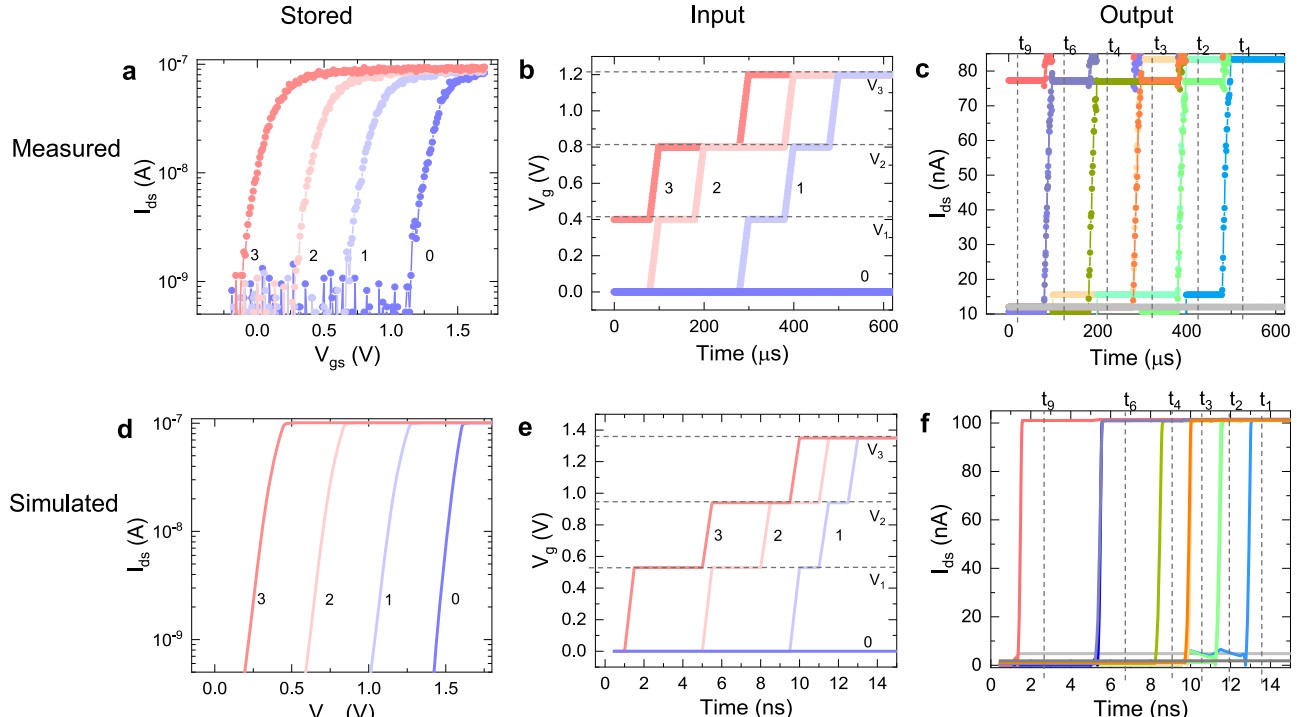

**Fig. 2 | Direct 2-bit multiply operation in a single cell. a** The measured $I_{ds}$-$V_{gs}$ characteristics of the 1FeFET-1R cell corresponding to the four stored states. **b** Input voltage against time applied to the FeFETs for measurement and (**c**) Output current against time on applying the input pulse. $I_{ds}$ rises at different instants of time corresponding to different outputs, which are used to get the product of input and stored states. **d**–**f** Stored, input, and output from a single cell multiplication operation are verified through simulations and determine the maximum speed of operation possible.

written to 4 distinct states, as shown in Fig. 2a. For details of the methods of writing the FeFET into the desired state, refer to the Methods section.

As stated earlier, for a single cell, the input is encoded in duration, and the magnitude of the voltage applied and shown in Fig. 2b. For an input of '0', the gate voltage is kept constant at 0 V. This voltage is less than the $V_{th}$ of any stored state in the FeFET, and no FeFET is turned ON. For the other inputs, the magnitude of the voltage is changed to levels $V_1$, $V_2$, and $V_3$ at a certain point in time, as stated in Eq. (1). The voltage level corresponds to the read voltage of the FeFETs storing '3', '2', and '1', respectively. Consequently, the FeFET storing '3' is turned on earliest, and a FeFET storing '0' is never turned ON. For input '3', voltage $V_1$ is applied at 0, and for input '1', voltage $V_1$ is applied later at 300 μs. Correspondingly, the FeFET storing '3' would turn on immediately for input '3' and turn on later for input '1'. Hence, the output of the single multiplication operation between the stored weight and the applied input is encoded as the time when the FeFET turns on, i.e.- the activation time of the FeFET.

Evaluation at different input conditions is performed from the resulting $I_{ds}$ – $V_{gs}$ transfer characteristics. The corresponding output is shown in Fig. 2c. The verification is done for a drain current of 75 nA. The timing marked for each output state is distinguishable. The current rises for an output of 9 first and at last for an output of 1. For intermediate output states, the activation time is in between. The commutative property is also maintained as seen for output of '2','3', and `6'. The instance of time when the current rises can distinguish between the different output states, as shown in Fig. 2c. This forms the basis for the MAC operation. It has to be noted that the slow readout observed in the experimental measurements is attributed solely to the limitations of the experimental setup. The measured data in Fig. 2c is post-processed from the transfer curves of Fig. 2a, hence the time is artificial and can be reduced further with direct measurements.

## Simulation

The functionality of the proposed in-memory macro is exemplified through simulations. The FeFET is simulated using a Preisach-based model[18] of the Ferroelectric capacitor and industry-standard compact model[19] of the underlying transistor (for details, see SI). We use 2-bit storage for the FeFET as in measurements, which corresponds to four different $V_{th}$ states. The simulation characteristics are matched to the experiments as closely as possible, and the resulting $I_{ds}$ – $V_{gs}$ characteristics are shown in Fig. 2d.

Correspondingly, we simulate the single-cell multiply operation as in measurements. The timing of the input pulse is modified such as to have almost linear characteristics of the output voltage against the desired output. Also, to determine the maximum speed of operation, the input pulse width is greatly reduced. The values are as per the fast readout proposed in FeFETs[5]. The input and the corresponding output are shown in Fig. 2e, f, respectively. Each output is distinguishable with no overlap, and also, the commutative property of the multiplication is maintained. We further simulated and evaluated the proposed MAC macro for a complete array of connected cells for neural network inference.

## Evaluation
### Crossbar level

The memory cells are arranged in the crossbar structure. In our recent work[20], we demonstrated that such an array can be programmed with MLC cells for up to 3b precision employing inhibit voltage levels and target erase schemes. A capacitor of 64 fF is connected at the bit-line for each column which is charged and discharged after every cycle. The value of the capacitor is chosen to allow the charging without being saturated. The voltage to which it is charged depends on the total current flowing into it and the time for which it flows, as given by equation Eq. (2). The time is determined by the input and stored state at which a particular FeFET activates. The

number of FeFETs activated at a given time determines the total current that flows into the capacitor. The voltage across the capacitor ($V_{cap}$) is sampled at a particular time ($t_{sampling}$) using a 2-bit ADC to get the final output.

The single cell connected to the capacitor is shown in Fig. 3a. The capacitor is charged with an almost constant current of 100 nA, and thus, $V_{cap}$ rises linearly with time as shown in Fig. 3b. The sampled voltage is maximum for the case of output 9 because $I_d$ turns on earliest in this case (at 1 ns) Fig. 3c. For the case of output 1, voltage is minimum because $I_d$ turns on the last (at 13 ns).

Similarly, in the case of 2 cells in the array, for output 18, $V_{cap}$ is maximum. Here, both the cells are activated at 1 ns, and the current is double that of one cell. Hence, the sampled voltage is also approximately double that of output 9. For output 1, only one of the FeFETs is turned on at 13 ns, and the sampled voltage is minimum. For intermediate values of output ('2','3','4', and '6') the sampled voltage at the capacitor lies in between the maximum and minimum as shown in Fig. 3c.

The reasoning can be extended to a higher number of cells in the column. The sampled voltage across the capacitor progressively increases with the number of cells activated. Figure 4 shows the sampled voltage ($V_{sampling}$) against the MAC output for up to 32 cells in the array. We considered all the possible input and stored combinations to generate the MAC output. We maintained a clear distinction between each output level and the numerical MAC output. However, with an increase in the number of cells, the linearity for higher output values is lost, and the sampling voltage starts to saturate. This is because, with an increase in the number of cells, the current in the source line increases which charges the capacitor faster. Alternatively, the equivalent resistance decreases and thus the time-constant decreases which saturates the capacitor voltage to its maximum value. This is also expected from equation Eq. (2). The capacitance value has a direct influence on the delay, which, in turn, affects the maximum speed of operation. Careful consideration is given to selecting the optimal capacitance value to prevent the sampling voltage from saturating while achieving the highest possible operational speed. However,

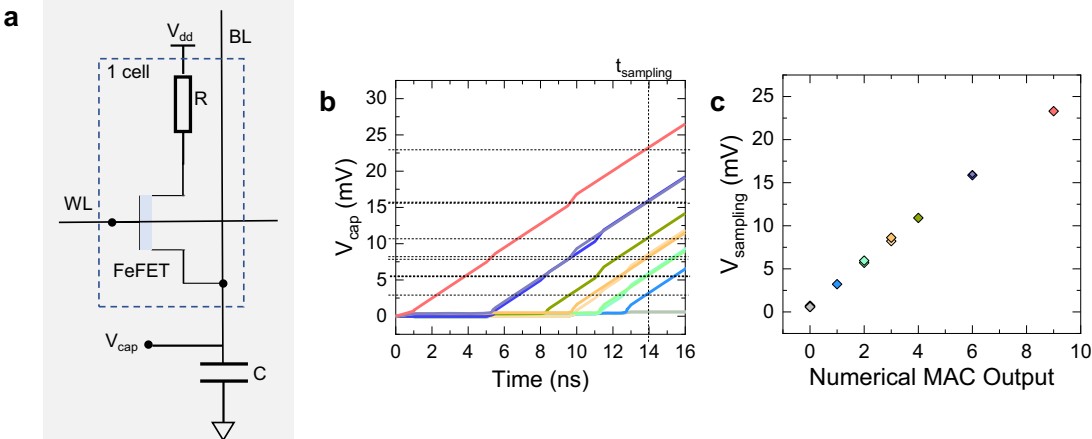

**Fig. 3 | Accumulation of the output as the voltage across the capacitor.**
**a** Structure of the single cell connected to the capacitor. **b** The voltage across the capacitor ($V_{cap}$) against time. As the current for output 9 (inp-3 × stored-3) is turned on first, the voltage across the capacitor at $t_{sampling}$ of 14 ns is highest. Similarly, for output 1 (inp-1 × stored-1), the voltage is lowest. **c** The sampling voltage across the capacitor ($V_{sampling}$) at $t_{sampling}$ vs the output MAC value. It is linear and perfectly aligned for the same output.

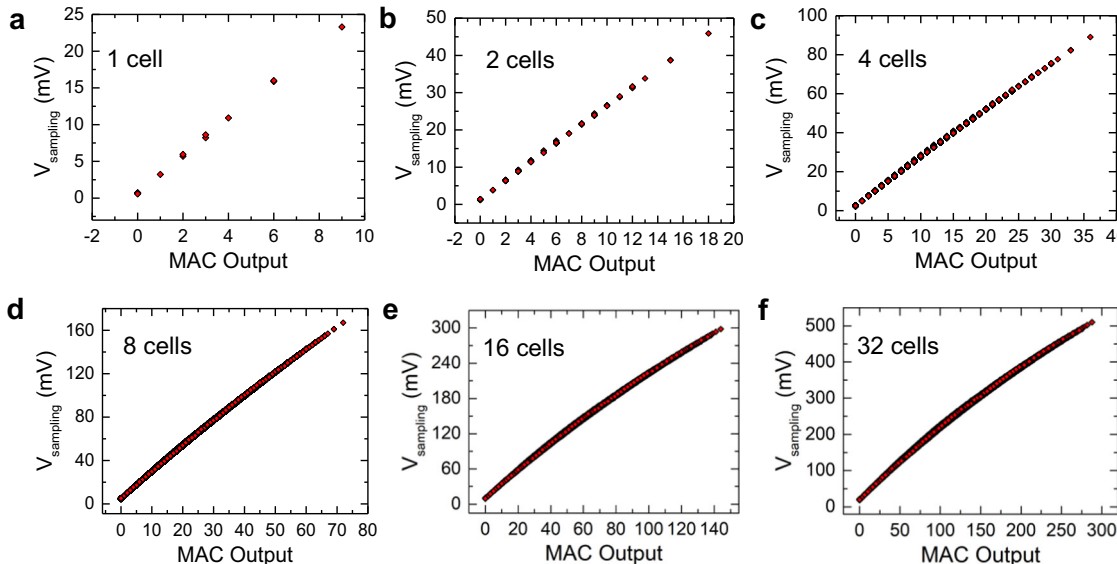

**Fig. 4 | Complete MAC output for increasing number of cells connected in the column.** **a**–**f** 1 cell to 32 cells sampled voltage across the capacitor against the MAC output connected in a single column. High degree of linearity is observed, which is desired for the final neural network inference.

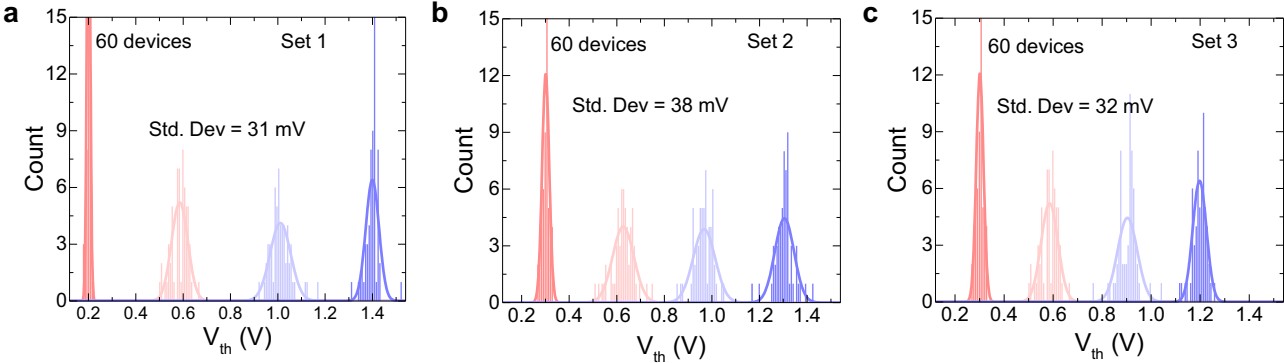

**Fig. 5 | Experimentally measured device-to-device variation with respect to $V_{th}$. a** Target condition 1. **b** Target condition 2. **c** Target condition 3. Different target conditions were set for measuring the standard deviation in target and measured $V_{th}$. A maximum standard deviation of 38 mV is observed.

achieving large capacitances can demand a substantial amount of area in the design layout. This introduces another trade-off between the speed of operation and the area required for the design. Balancing these trade-offs becomes crucial in designing efficient systems that achieve both desirable operational speed and occupy a reasonable area footprint.

The final output is converted into 2 bits using strong-ARM comparators[15] connected across the capacitor. The comparasion threshold levels are selected based on the quantization required. For the final simulations of the DNN, 32 cells in a column are considered. The crossbar is divided into 32 × 32 cells in a segment. Each column in the segment is connected to the 2-bit ADC. The quantized weights are directly written to the FeFET crossbar (For details on the quantization, see SI). The quantized input is applied using a digital-to-analog converter (DAC) connected to the word line for each row, which selects a particular voltage based on the applied input.

Afterwards, the variability in the FeFET devices are incorporated in the FeFET model to calculate the loss in inference accuracy of the neural network. The variation in $V_{th}$ is measured from real fabricated FeFET devices. Figure 5 shows the distribution for three different target condition sets of $V_{th}$. In the first set, the target levels are chosen as 0.2 V, 0.6 V, 1.0 V and 1.4 V. In the next set, the target levels are 0.3 V, 0.633 V, 0.967 V and 1.3 V. In the third set, the target levels are 0.3 V, 0.6 V, 0.9 V and 1.2 V. The maximum standard deviation of 38 mV is obtained for the difference in actual and target $V_{th}$.

For simulations, a standard deviation of 40 mV for $V_{th}$ of the FeFET is assumed for each state. 1000 Monte Carlo samples for each stored, and the input value is simulated for up to 4 cells in the array. For a higher number of cells, the total variability of each state is calculated algebraically from the lower number of cells (for details, see SI). A maximum standard deviation of less than 4 mV for any given output state is observed in the case of 32 cells in the array. Quantization of the output into 4 levels (2 bits) further reduces the error probability. Finally, the neural network models are simulated to calculate the loss in inference accuracy and derive the performance metrics of the proposed MAC macro.

## Architecture level

We used the experimental and simulation data to estimate the performance of the demonstrated macro on two neural network models. We perform the inference of these networks on the target macro and at limited variability of 40 mV based on our measurements. As shown in Fig. 6a, b, we fully quantize the LeNet model[21] for MNIST[22] into 2-bit activation and 2-bit weights to fit our macro capabilities of MAC precision. The network consists of three convolutional layers and two dense layers. The network requires 397920 MAC operations using 61610 parameters. Considering a max of 40 mv device

variation, we achieve 96.64% network accuracy compared to the original model accuracy of 99.11% achieved at full floating point precision.

Additionally, we also quantize two layers from the VGG19[23] network for the CIFAR-10[24] dataset as shown in Fig. 6c, d. The network consists of 19 layers. However, we tested and quantized only two convolutional layers. The networks require 38947914 parameters out of which we quantized only 1179648 parameters to 2-bit, where the rest are quantized to 8-bit. The 2-bit quantization layers use 2-bit quantized activations. We considered a max of 40 mv device variation for those two layers. We achieved 91.55% network accuracy compared to the original model accuracy of 93.22%. Nevertheless, we tested the network's accuracy in case of smaller device variation. Accuracy of 97.25% and 91.9% for LeNET and VGG19 was observed for the case of 30 mV variation in $V_{th}$. However, for 25 mV and lower, there is no overlap and hence we achieve full software equivalent accuracy.

As demonstrated, the variation has a very limited influence on the accuracy of the final network resulting in less than 4% in both networks over the testing set. This can be reasoned by the cell architecture of 1FeFET-1R, which limits the current and the quantization of the final output into 2 bits, which limits the impact of the variation on the sampling voltage thresholds. Also, the device variation has less error probability on the states corresponding to the values 2 and 3 compared to the values 0 and 1 as shown in Fig. 5. This error distribution is reflected in the accuracy as the least significant bit has a lower impact on the network accuracy compared to the most significant bit.

To further demonstrate the reliability of our proposed system, we analysed the impact of retention and endurance of the device on the tested network's accuracy. Extrapolated 10-year retention and endurance of $10^{10}$ cycles have been demonstrated for FeFET in the literature[25–27]. The retention and endurance characteristics of our MLC FeFET are shown in Fig. 7. We observed minimal changes in the threshold voltage state, and also minimal overlap among the states over the entire studied time span ($10^5$ s). This results in no further accuracy loss in the case of LeNET while for VGG19 accuracy drops to 90.67% (from 91.9%) after $10^5$ s. The endurance of our FeFET cell under the target programming and verify scheme is at least $10^4$ cycles for the devices in the array. The usage of the write-verify algorithm which continuously writes until the required target or the maximum pulse count is reached accelerates degradation. Proper selection of the targets could increase endurance at the cost of a reduced memory window. Also, a better write algorithm that detects the target value won't be reached anymore after a few tries and therefore limiting the write voltage from there on could improve endurance. However, the focus of this work is only neural network inference (i.e - no training), where the memory states are written once at the beginning and very rarely after that, there is no drop in accuracy resulting from the endurance.

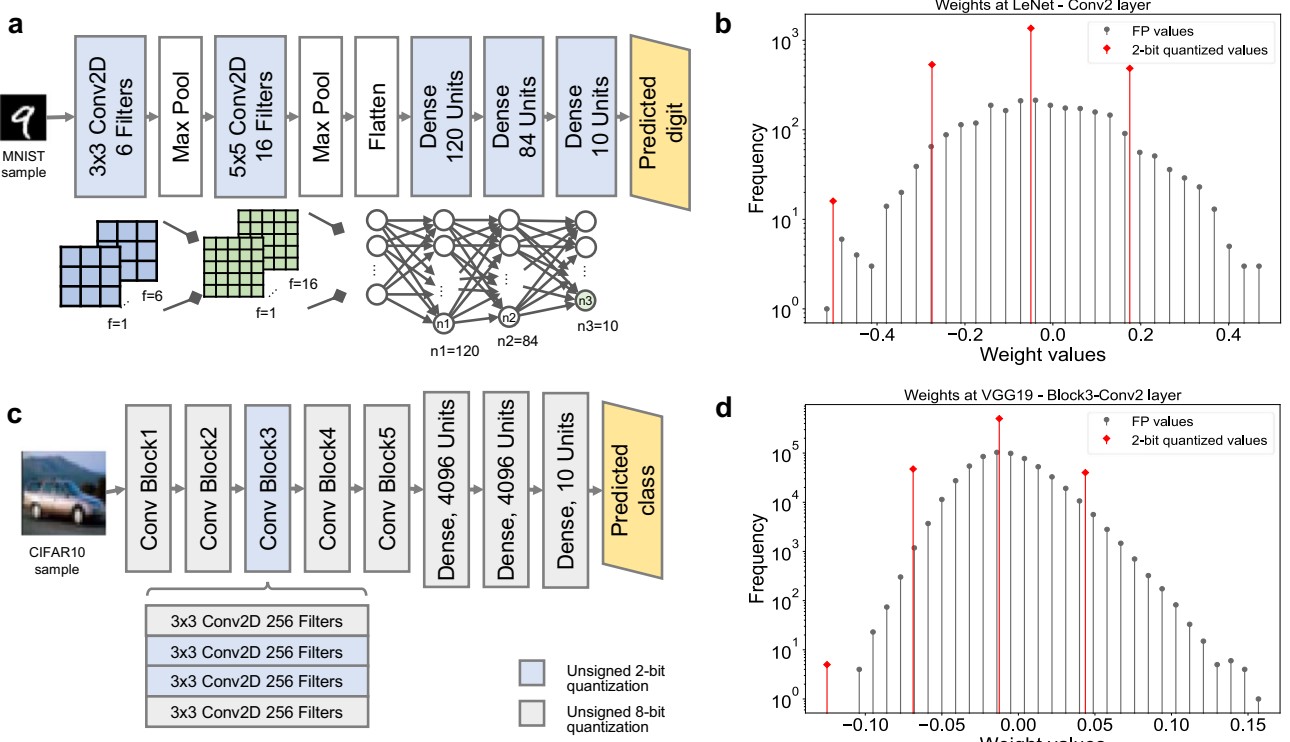

**Fig. 6 | Neural network simulation utilizing the proposed IMC MAC macro.**
**a** LeNet neural network is tested for handwritten digit recognition MNIST dataset. All the MAC layers are quantized to 2 bits. **b** The weight distribution shows the trained weights when they are re-quantized to 2 bits for the second convolutional layer. An accuracy of 96.64% is achieved considering device variations. **c** VGG19

neural network is tested for object classification CIFAR-10 dataset where only two convolutional layers are quantized for testing. **d** The weight distribution shows the trained weights when they are re-quantized to 2 bits for a convolutional layer in VGG19. An accuracy of 91.55% is achieved under the effects of device-to-device variations.

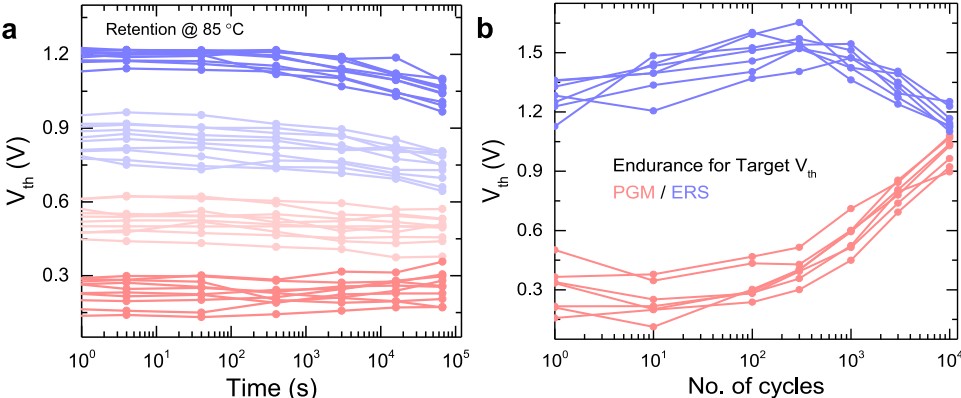

**Fig. 7 | Experimentally measured retention and endurance of the FeFET cells with respect to V_th. a** 24-h retention characteristics at 85 °C of the four target V_th states of the FeFET showing almost steady retention for the entire duration.

**b** Endurance of the proposed cell under subsequent target program-erase cycles (standard erase followed by target programming by increasing pulse amplitude and verify of V_th) with a pulse width of 200 ns.

In our experiments, the crossbar and the ADCs consume 153.6 μW measured directly for the given frequency(66 MHz) and accumulation capacitance (64 fF). The energy efficiency of the presented crossbar accordingly is 885.4 tera-operations per second power watt (TOPS/W), where each operation refers to 2-bit/2-bit multiply or accumulate. We compared our design against existing in-memory crossbars which are shown in Table 1.

## Discussion

In this work, we have demonstrated the multi-bit MAC operation exploiting for the first time a multi-bit FeFET cell and a novel encoding

and decoding scheme. The variability was controlled with the help of the external current-limiting resistor approach. Compared to other works in the literature with different memory architectures, we show a higher throughput and efficiency. The results portray that FeFET can be a strong contender for DNN acceleration with high efficiency.

We see an important potential for our computational innovation to allow for a dense IMC macro that can use simple memory cell 1FeFET-1R while performing multi-bit MAC operation per each cell. Additionally, the combination of several computational dimensions time/voltage/current encodings allow for combating the variability and maintaining the macro efficiency in an unprecedented way.

**Table 1 | Comparison of in-memory crossbars with this work**

| Crossbar | Memory | Tech. [nm] | Freq. [MHz] | Power [µW] | Throughput [GOPS] | Efficiency [TOPS/W] |
|---|---|---|---|---|---|---|
| ISSCC'22[33] [a] | SRAM | 28 | 333 | - | - | 438 |
| SLC-MLC[34] | PCM | 40 | 307 | 14900 | 3900 | 261 |
| nvCIM[35] | RRAM | 55 | 1 | 0.0322 | 0.002 | 62.11 |
| Samsung[36] | MRAM | 28 | 11.1 | 225 | 91.125 | 405 |
| This work | FeFET | 28 | 66 | 153.6 | 136 | 885.4 |

[a]Performance scaled to 2-bit/2-bit precision.

## Methods

### 1FeFET-1R cell characterization

To write the FeFET, into 4 distinct states (i.e 2-bit), we employed the write-verify scheme. A FeFET is written into a fully programmed state by applying +4.5 V for 500 ns and into a fully erased state by applying −5 V for 500 ns while keeping the source and drain terminals grounded. Before evaluation, each FeFET is cycled 50 times with these conditions for preconditioning. A FeFET is then written to 4 distinct states using a write-verify-scheme. Therefore, the FeFET is initially fully erased[28,29]. Starting with a write voltage of 1.4 V for 200 ns, the FeFET is gradually programmed. Write voltage is incremented in steps of 40 mV. After each write pulse, a delay of 500 ms is applied for charge detrapping, and a read operation verifies the state. This scheme is continued until the target value is reached. Target levels are selected at 0.3 V, 0.7 V, 1.1 V, and 1.5 V at a constant current condition of 80 nA. After setting the target state, a final readout is performed. Before reading, a sufficiently large time is waited for any charge detrapping. In this case, a delay time of 2 seconds is chosen. For reading a voltage ramp, $V_G$ from −0.2 V to 1.7 V in steps of 10 mV is applied to the gate. Current $I_D$ is measured at the drain side while biasing the drain-terminal at 0.1 V, obtaining the 4 distinct $I_{ds} - V_{gs}$ curves.

For the retention measurements at 85 °C shown in Fig. 7a, a pulse width of 400 ns is used to measure the $V_{th}$ of each state at different intervals of time for up to $10^5$ s. The states are written using the "write-verify" scheme to set it to a particular $V_{th}$. For a more detailed description see SI. For the endurance measurements shown in Fig. 7b with target programming for MLC first, erase pulse is applied with a magnitude of −5 V followed by increasing magnitude pulses of duration 200 ns and verify of $V_{th}$ until the required state is reached or the maximum number of pulses or voltage is detected. This represents the used target-verify scheme of programming the FeFET to a desired state. This cycle is repeated for up to $10^4$ cycles.

### Simulation methodology

All the simulations are performed in the commercial SPICE simulator Cadence Spectre. For simulating the FeFET, a Preisach-based model of the FeFET is considered along with the BSIM-IMG model of the transistor[18,19]. To simulate the FeFET crossbar array, a single column in the crossbar is first selected. Each cell in the crossbar is a 1FeFET-1R structure. The ADC connected to the column is simulated using well-calibrated BSIM-IMG transistors based on measured data[30]. For all possible input and stored combinations, netlists for the column are generated using a Python script. The netlists are then run in SPICE, and the results are extracted. To include the variability of the FeFET on the MAC output, Monte Carlo simulations are performed following a normal distribution with $3\sigma$ truncation. 1000 sample Monte-Carlo runs is simulated for each possible input and stored combination. The mean, 5th, and 95th percentile of the sampling voltage are extracted corresponding to each MAC output. Finally, the output is sampled using the ADC. The threshold voltages of the comparators are chosen according to the 2-bit quantization of the neural network. Finally, the output levels are converted to binary values and are used for the neural network simulation.

To assess the impact of FeFET variability and the DNN quantization on the performed task accuracy, a bit-accurate simulation of the multi-level FeFET macro was implemented using the simulation framework ProxSim[31] based on Tensorflow[32]. We implemented a custom CUDA operator to simulate the results from different simulations and measurements.

## Data availability

The data generated in this study and used in the figures and plots within this manuscript have been deposited in the public database repository under: https://github.com/TUM-AIPro/Nature Communications_CiM_FeFET.

## Code availability

The codes that supports the findings of this study has been deposited in the public database repository under: https://github.com/TUM-AIPro/NatureCommunications_CiM_FeFET.

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

## Acknowledgements

This work has received funding from the ECSEL Joint Undertaking (JU) under grant agreement No 826655 and No 876925. The JU receives support from the European Union's Horizon 2020 research and innovation programme and Belgium, France, Germany, Portugal, Spain, The Netherlands, Switzerland.

## Author contributions

T.S., S.C., and H.A. conceived the idea. T.S. and S.C. contributed equally to the work. H.A., Y.C. supervised the analysis at the device and circuit levels. N.W. supervised the analysis at the architecture level. T.Ki supported T.S. in his analysis. T.Kä supervised the experimental demonstration. F.M. performed the circuit/device measurements. S.C., T.S., and N.L. conducted the device and circuit simulations, variation analysis, and architectural-level benchmarking. All authors contributed to the manuscript writing and provided feedback.

## Funding

## Competing interests

The authors declare no competing interests.
