## [Peer Review File · Nature Communications]

REVIEWER COMMENTS

Reviewer #1 (Remarks to the Author):

The authors are demonstrating the application of ferroelectric field-effect transistors for in-memory computing operation utilizing multi-bit multiply and accumulate operations. The work is very detailed and rigorous when it comes to analysis of the computing architecture. The main thrust of the work is to demonstrate the feasibility of the computing approach and its realization in FAB-grown devices. If we consider the traditional co-design stack, the level of this work is somewhere between devices and algorithms, which assumes that the fundamental materials and devices physics is already well understood and incorporated into compact models. I therefore believe it is too device oriented for the scope of Nature Communications and should be considered for a more specialized journal.

Reviewer #2 (Remarks to the Author):

Ferroelectric technology is known to be a promising NVM candidate for memory and AI applications. This work provides comprehensive studies to implement novel 1FeFET-1R unit cell based crossbar array for inference applications, and this makes this work potentially very valuable to the research community. However, in order to meet the high standards of Nature Communications, this work still needs to be further improved. Since this paper targets inference applications, it would be important to discuss more on read latency to the array as well as the V_t shift over time under MLC (i.e., retention in partially programmed states) on the FeFETs. These two topics are still lacking in the manuscript, and therefore in revised manuscript, it would be useful to provide more discussions on the improvement strategy or clarification of the limit from 1FeFET-1R unit cell based crossbar array regarding these two aforementioned aspects. I have below detailed comments.

1) Comparing to typical memory application (i.e., just 1 and 0 by set and reset operation), it has been observed that the 2 bit MLC posts more challenges to NVM technology to meet more strict V_t shift requirements [Ref1]. Although FeFET shows good potential to have 10 year extrapolated retention for a typical memory application [Ref2], it is not clear if MLC for FeFET in this work would show sufficiently stable V_t over time for all of the 0, 1, 2, 3 levels within the array, and also if such V_t shift would still be at a tolerant level for the inference application (e.g. acceptable accuracy loss on MNIST, CIFAR-10 and beyond). Of particular, it is especially important to evaluate V_t shift in partially programmed states (i.e., 1, and 2) compared to fully programmed and erased states (i.e., 0, and 3). It is important to include some thoughts in that perspective and discuss if the V_t shift with time on the 2 bit MLC in FeFETs would be a concern in the large array for inference application.

2)The other important aspect for inference application is read latency. In figure 2 of the manuscript, simulation work provides good potential for < 20ns time scale, however, experiments on FeFET array needs 600 us time scale. The big difference between experiment and simulation work is not clearly discussed in details in the manuscript, and it casts doubt on the cause of such slow read from experiments and its potential impact on inference tasks. Would that due to experimental set up (e.g. probe card, or test structure) or because of the proposed unit cell that includes 1M-ohm resistor, or related to the capacitor from the decoder? Most importantly, is this slow read related to the 32x32 FeFET based array and if that is a big concern in even larger array? If so, would that be an issue when larger FeFET array is used for inference application using larger NN? It would be important to discuss the 600us read latency in the experiments with improvement strategies listed in revised manuscript. If this slow read is due to serve trapping after programming, which has to require enough detrapping time in order to get enough memory window, then it is vital important to include V_t dynamics within 100us time scale, in order to clarify the origin of read latency and provides future direction for technology improvement.

3)One additional comment, the LeNet on MNIST shows test accuracy 96.64% when using FeFETs that have 40mv device to device variation, would this test accuracy improve significantly if device to device variation would be further improved a little bit? If this could be evaluated by some simulation work, that would be a very helpful guideline for ferroelectric technology improvement.

Ref1. N. Gong, et al., "A No-Verification Multi-Level-Cell (MLC) Operation in Cross-Point OTS-PCM", VLSI, 2020. DOI: 10.1109/VLSITechnology18217.2020.9265020

Ref2. T.P. Ma, et al., "Retention and Endurance of FeFET Memory Cells", IMW, 2019. DOI: 10.1109/IMW.2019.8739726

Reviewer #3 (Remarks to the Author):

In the manuscript, the authors employ the 1FeFET-1R cell structure for MAC driving and present a crossbar macro with minimal cell-to-cell variability effects. The authors utilize the MLC FeFET device and achieve 96.6% accuracy in handwriting recognition and 91.5% accuracy in image classification through simulations. However, additional experimental results are necessary to validate the FeFET device discussed in this manuscript. For this manuscript to be published in Nature Communications, a major revision is required, and the specific details are as follows:

1. Figure 1b illustrates the variation in threshold voltage of the transfer curve caused by the multi-state FeFET. However, the distinction between threshold voltages for states 0 and 1 is not clearly discernible from states 2 and 3. This is a crucial concern that diminishes the reliability of information storage when using FeFET in an array format. In order to clearly distinguish the state between 0 and 1, please add the distribution of threshold voltage according to each information state.

2. The authors primarily focus on demonstrating device-to-device reliability due to the array's characteristics. However, to utilize the memory device FeFET, measuring the device's endurance and retention is essential. This is very important as an index showing the possibility of a device being applied to an actual circuit. Considering the FeFET device's utilization of multi-state states, it is necessary to provide evidence of endurance and retention for each multi-state.

3. Additional clarification is needed to explain the reduction in power consumption resulting from including 1R, as mentioned in lines 118 to 120 on page 8.

4. In lines 228 and 231 of page 16 of the manuscript, it is observed that the linearity of the V_{sampling} value for MAC output diminishes as the number of cells increases. The authors attribute this phenomenon to Equation 2; however, additional explanation is warranted to further clarify the application limitations of the proposed IMC crossbar macro. Please ensure a more detailed explanation of this observation, addressing the reasons behind the loss of linearity in the V_{sampling} value with an increasing number of cells.

Response Letter and List of Updates

First Demonstration of In-Memory Computing Crossbar using Multi-level Cell FeFET

Nature Communications, <NCOMMS-23-21473>

– Major Revision –

We would like to thank the editors for arranging the review of our manuscript as well as all reviewers for their valuable comments, which have helped in improving the quality of the manuscript. We provide a point-by-point response to their questions below in blue color. Changes in the manuscript are explicitly mentioned here and highlighted in the revised manuscript. Please note that all changes are marked also in blue color. Reference, figure, and other numbers associated with this response letter are preceded by the letter “R”.

* Summary of Major Changes

1. We investigated and reported from device measurements the retention and endurance of the multi-level cell device.
2. We performed new experiments to demonstrate the impact of variations and retention on the inference accuracy of the tested neural networks.
3. We revised and extended several parts in the main text to clarify several points raised by the reviewers related to the scope of our work, the measurements setup, and power consumption.

Reviewer 1

RIQ1 - The authors are demonstrating the application of ferroelectric field-effect transistors for in-memory computing operation utilizing multi-bit multiply and accumulate operations. The work is very detailed and rigorous when it comes to analysis of the computing architecture. The main thrust of the work is to demonstrate the feasibility of the computing approach and its realization in FAB-grown devices. If we consider the traditional co-design stack, the level of this work is somewhere between devices and algorithms, which assumes that the fundamental materials and devices physics is already well understood and incorporated into compact models. I therefore believe it is too device oriented for the scope of Nature Communications and should be considered for a more specialized journal.

RIA1 - We would like to thank the reviewer for the effort and positive assessment. The key goal of this work is to demonstrate a *multidisciplinary* approach that starts from the device level and going through circuit level and architectural level all the way up to algorithmic level. For that reason, we considered Nature Communications which positions itself as a multidisciplinary journal.

Our core contributions target circuit and architecture levels while leveraging the FeFET devices to achieve the above-mentioned goals. We demonstrate in this work how the unique characteristics of FeFET devices can be employed to perform 2-bit multiply-accumulate (MAC) operations, for the first time, leading to unprecedented efficiency improvement for deep learning applications.

To demonstrate better the interdisciplinary nature of our work, we re-structured the manuscript in which we first highlight the FeFET device properties and deployment of our idea on the device level. Then, we elaborate on the structure of the crossbar and how it can perform the MAC operations. The results and evaluation section show the applicability of our work to different crossbar sizes using measurements and simulations and considering device related variations. Finally, we evaluate the impact of the device-to-device variations, endurance and retention characteristics of the device on the algorithmic level by testing two applications and showing the minimal impact we have on their accuracy while achieving unprecedented computational efficiency.

Changes in the revised manuscript: In the revised manuscript, we have highlighted the multidisciplinary nature of our work and illustrated the contributions at device, circuit, architecture, and algorithmic levels. The new text is added to the manuscript in pages 7 and 8 from line 106 to 110.

Reviewer 2

Ferroelectric technology is known to be a promising NVM candidate for memory and AI applications. This work provides comprehensive studies to implement novel 1FeFET-1R unit cell based crossbar array for inference applications, and this makes this work potentially very valuable to the research community. However, in order to meet the high standards of Nature Communications, this work still needs to be further improved. Since this paper targets inference applications, it would be important to discuss more on read latency to the array as well as the V_t shift over time under MLC (i.e., retention in partially programmed states) on the FeFETs. These two topics are still lacking in the manuscript, and therefore in revised manuscript, it would be useful to provide more discussions on the improvement strategy or clarification of the limit from 1FeFET-1R unit cell based crossbar array regarding these two aforementioned aspects. I have below detailed comments.

We would like to thank the reviewer for the effort and encouraging comments. In the following section, we provide a detailed discussion of every comment raised by the reviewer.

R2Q1 - Comparing to typical memory application (i.e., just 1 and 0 by set and reset operation), it has been observed that the 2 bit MLC posts more challenges to NVM technology to meet more strict V_t shift requirements [Ref1]. Although FeFET shows good potential to have 10 year extrapolated retention for a typical memory application [Ref2], it is not clear if MLC for FeFET in this work would show sufficiently stable V_t over time for all of the 0, 1, 2, 3 levels within the array, and also if such V_t shift would still be at a tolerant level for the inference application (e.g. acceptable accuracy loss on MNIST, CIFAR-10 and beyond). Of particular, it is especially important to evaluate V_t shift

in partially programmed states (i.e., 1, and 2) compared to fully programmed and erased states (i.e., 0, and 3). It is important to include some thoughts in that perspective and discuss if the V_t shift with time on the 2 bit MLC in FeFETs would be a concern in the large array for inference application.

Ref1. N. Gong, et al., “A No-Verification Multi-Level-Cell (MLC) Operation in Cross-Point OTS-PCM”, VLSI, 2020. DOI: 10.1109/VLSITechnology18217.2020.9265020

Ref2. T.P. Ma, et al., “Retention and Endurance of FeFET Memory Cells”, IMW, 2019. DOI: 10.1109/IMW.2019.8739726

R2A1 - We thank the reviewer for raising this important concern regarding the retention of MLC. We completely agree with the reviewer on the importance of analyzing the DNN accuracy loss under the retention effect. Recent studies have demonstrated stable threshold voltage (V_{th}) of the HfO₂-based FeFET cells for the entire time studied, not only for conventional one-bit memory cells but also for multi-level cells [R2.1, R2.2]. As requested by the reviewer, we have now investigated the retention characteristics of the proposed multi-level cell and the corresponding impact on DNN's inference accuracy. The measured retention behavior of the 2-bit cell is illustrated in Fig. R2.1, which demonstrates the V_{th} drift over time at two different temperatures (25 °C and 85 °C). The randomly distributed states (without verify) retention measurements at 25 °C are carried out to understand the evolution of the V_{th} over time. The higher V_{th} states drift towards lower values whereas the lower V_{th} states remain fairly constant. Fig. R2.1(b) shows the target V_{th} retention measurements of four particular states set with the write-verify scheme at 85 °C.

Figure R2.1: Measurements for retention characteristics of the proposed 2-bit per cell FeFET.

(a) Retention at 25 °C for random V_{th} states without target programming (b) Retention at 85 °C for target V_{th} states.

Figure R2.2: Distribution of the threshold voltage states at (a) $t=0$ s (i.e., at the beginning of the lifetime) and at (b) $t=10^5$ s (i.e., in the scope of the V_{th} drifts induced by the retention degradation) showing the minimal overlap between the states.

As can be observed, despite the drift in V_{th} , the safety margin among the four stored states remains sufficient. The downward drift for the high threshold voltage states leads to a slight increase in the overlap among the states observed over the entire studied time span. To quantify the consequence of such a small increase in the overlap among the different states stored within the FeFET device, we performed additional experiments to determine the loss in DNN accuracy. The overlap in the V_{th} distribution is extrapolated for the required states from the measured data from Fig. R2.1 and shown in Fig. R2.2. The overlap of the memory states translates to an overlap in the MAC output from the array and finally an error in the DNN's inference. The new analysis shows that drift due to FeFET retention results in merely a 0.8% loss in accuracy (from 91.55% to 90.67%) in the case of VGG19 whereas for LeNET the accuracy remains constant. Hence, the small overlap in the memory states does not cause an observable error at the algorithmic level.

Changes in the revised manuscript: In the revised manuscript, we have added a discussion regarding the impact of retention. We have added the retention measurements for our FeFET MLC in Figure 6a. We also reported the impact of threshold shift on the DNN inference accuracy using our proposed method. The new text is added to the manuscript from lines 300 to 307 and 352 to 358. Figure 6 is added to the main text. Figure S6 is added to the supplementary information.

R2Q2 - The other important aspect for inference application is read latency. In figure 2 of the manuscript, simulation work provides good potential for < 20 ns time scale, however, experiments on FeFET array needs 600 us time scale. The big difference between experiment and simulation work is not clearly discussed in details in the manuscript, and it casts doubt on the cause of such slow

read from experiments and its potential impact on inference tasks. Would that be due to experimental set up (e.g. probe card, or test structure) or because of the proposed unit cell that includes 1M-ohm resistor, or related to the capacitor from the decoder? Most importantly, is this slow read related to the 32x32 FeFET based array and if that is a big concern in even larger array? If so, would that be an issue when larger FeFET array is used for inference application using larger NN? It would be important to discuss the 600us read latency in the experiments with improvement strategies listed in revised manuscript. If this slow read is due to serve trapping after programming, which has to require enough detrapping time in order to get enough memory window, then it is vital important to include V_t dynamics within 100us time scale, in order to clarify the origin of read latency and provides future direction for technology improvement.

R2A2 - We acknowledge and agree with the reviewer's observation. As accurately pointed out by the reviewer, the slow read-out in the experimental results primarily stems from limitations in the measurement setup due to the impact of the PXIe system and the probe station as highlighted in supplementary material S2, which introduce significant delays. Further, the measurement of the multiplication output is post-processed from the transfer curves and hence the timing is artificial and represents the time after which the voltage is reached. Direct measurements can yield a smaller time scale, however, limitations in the experimental setup prevent us from that. Importantly, it has been recently demonstrated that FeFET can have a very fast read-out of the memory states in the order of nanoseconds either by using de-trapping pulse once after writing or gate stack engineering [R2.3, R2.4]. Since we target inference-only for DNN applications, we can assume the device has been already written prior to its deployment (i.e., at the setup time). During the run-time only read

operations are performed as long as the weight parameters of the deployed neural network remain unchanged.

The actual delay for the multiply and accumulate (MAC) operation (and thereby the frequency of operation) is constrained by the capacitance associated with the decoder and the resistor connected to each cell and the speed of the comparator. However, the “strong-arm” latch employed in this work can have a very high frequency of operation and is, therefore, not the limiting factor [R2.5]. While a larger array size introduces an additional capacitance on the bit lines, it is worth noting that the external capacitor connected to the decoder plays the prominent role in determining the readout delay.

In the case of larger arrays, the sampling voltage begins to saturate. This phenomenon becomes apparent through the deviation from linearity observed when a higher number of cells (such as 16 cells and 32 cells) are present in the column, as depicted in Fig. R2.3. As the number of cells further increases, a larger capacitance is required, consequently leading to increased delays. Therefore the value of the capacitance is carefully chosen so as not to saturate the sampling voltage and yet have the maximum speed of operation.

In order to address the reviewer’s question regarding charge trapping and detrapping, we would like to clarify that there is an initial detrapping time necessary for the read operation following a write operation. However, since the focus is on inference applications, where the FeFET devices are written only once to store the necessary parameters of the DNN (i.e the weights of the network), and then the multiple MAC operations are repeatedly performed thereafter.

Figure R2.3: The evolution of the sampling voltage while going from 1 cell in the array to 32 cells in the array shows the degradation of linearity. The connected capacitor is 64 fF.

To summarize, the slow readout observed in the experimental results is attributed to the limitations of the experimental setup. In general, FeFET devices can exhibit fast read-out even in arrays and can be utilized for high-throughput operations. The focus of our work is DNN inference (i.e. no training) which necessitates writing the FeFET devices only once to store the weights parameters and then frequent reading of the devices to execute MAC operations.

Changes in the revised manuscript: In the revised manuscript, we have thoroughly added the reason for the high time scale in the measured data. We have extensively discussed how the array size and the value of the capacitance are interrelated and their influence on the overall delay of the MAC operation. The new text is added to the manuscript from lines 189 to 192 and lines 237 to 249.

R2Q3 - One additional comment, the LeNet on MNIST shows test accuracy 96.64% when using FeFETs that have 40mV device to device variation, would this test accuracy improve significantly if device to device variation would be further improved a little bit? If this could be evaluated by some simulation work, that would be a very helpful guideline for ferroelectric technology improvement.

R2A3 - In this work, we have considered a device-to-device variation of 40 mV which had a small impact on the inference accuracy leading to 96.6% and 91.5% for LeNET and VGG19, respectively. This translates to less than 1% accuracy loss compared to the golden scenario where mature CMOS technology is used. For 30 mV variation in V_{th} of the FeFET device, accuracy of 97.25% and 91.9% is calculated for the LeNET and VGG19, respectively. For values less than 30 mV variation in V_{th} , no loss in inference accuracy is observed compared to the golden scenario. This gives the guidelines to which the device variations must be restricted in order to have no loss in DNN inference accuracy.

Changes in the revised manuscript: In the revised manuscript, we have added the discussion on how a lower V_{th} spread affects the inference accuracy. The new text is added to the manuscript from lines 288 to 292.

References

- [R2.1] F. Müller, S. De, R. Olivo, M. Lederer, A. Altawil, et al., “Multilevel operation of ferroelectric fet memory arrays considering current percolation paths impacting switching behavior,” IEEE Electron Device Letters, vol. 44, no. 5, pp. 757–760, 2023.

- [R2.2] B. Zeng, M. Liao, Q. Peng, W. Xiao, J. Liao, et al., “2-bit/cell operation of $\text{Hf}_{0.5}\text{Zr}_{0.5}\text{O}_2$ based FeFET memory devices for NAND applications,” *IEEE Journal of the Electron Devices Society*, vol. 7, pp. 551–556, 2019.
- [R2.3] H. Mulaosmanovic, E. T. Breyer, S. Dünkel, S. Beyer, T. Mikolajick, et al., “Ferroelectric field-effect transistors based on HfO_2 : A review,” *Nanotechnology*, vol. 32, no. 50, p. 502002, 2021.
- [R2.4] M. Hoffmann, A. J. Tan, N. Shanker, Y.-H. Liao, L.-C. Wang, et al., “Fast read-after-write and depolarization fields in high endurance n-type ferroelectric FETs,” *IEEE Electron Device Letters*, vol. 43, no. 5, pp. 717–720, 2022.
- [R2.5] B. Razavi, “The strongarm latch [a circuit for all seasons],” *IEEE Solid-State Circuits Magazine*, vol. 7, no. 2, pp. 12–17, 2015.

Reviewer 3

In the manuscript, the authors employ the 1FeFET-1R cell structure for MAC driving and present a crossbar macro with minimal cell-to-cell variability effects. The authors utilize the MLC FeFET device and achieve 96.6% accuracy in handwriting recognition and 91.5% accuracy in image classification through simulations. However, additional experimental results are necessary to validate the FeFET device discussed in this manuscript. For this manuscript to be published in Nature Communications, a major revision is required, and the specific details are as follows:

We would like to thank the reviewer for the effort. We agree with the reviewer that more experiments can strengthen our contributions and completely validate our work. We detail our reply and the changes in the manuscript in the following point-to-point response.

R3Q1 - . Figure 1b illustrates the variation in threshold voltage of the transfer curve caused by the multi-state FeFET. However, the distinction between threshold voltages for states 0 and 1 is not clearly discernible from states 2 and 3. This is a crucial concern that diminishes the reliability of information storage when using FeFET in an array format. In order to clearly distinguish the state between 0 and 1, please add the distribution of threshold voltage according to each information state.

R3A1 - We thank the reviewer for pointing out this issue. We have now extracted and added the threshold voltage distribution corresponding to the transfer curves. The new data is presented in Fig. R3.1. As can be observed, all the stored states are discernible with very minimal overlap. The

Figure R3.1: **Distribution of the threshold voltage corresponding to the four states.**

chosen threshold voltage (V_{th}) values for the $I_{ds} - V_{gs}$ curves shown are 0.2 V, 0.6 V, 1 V, and 1.3 V. Correspondingly for any set of distinguished and evenly placed threshold voltage states, we can have a sufficient separation as shown in Figure 5 of the main manuscript for three more sets of measurement for different target V_{th} . Note that in our analysis at the architectural and algorithmic levels, we take the impact of variations and thus the overlap between states into account and translate it to the corresponding error in the MAC operation. This is later translated to the corresponding loss in neural network accuracy.

Changes in the revised manuscript: In the revised manuscript, we have added the specific distribution in Figure 1 and discussed how the overlap affects accuracy in lines 297 to 300.

R3Q2 - The authors primarily focus on demonstrating device-to-device reliability due to the array's characteristics. However, to utilize the memory device FeFET, measuring the device's endurance

and retention is essential. This is very important as an index showing the possibility of a device being applied to an actual circuit. Considering the FeFET device's utilization of multi-state states, it is necessary to provide evidence of endurance and retention for each multi-state.

R3A2 - We completely agree with the reviewer. Recently, multi-level-cell FeFET with excellent endurance (upto 10^{10} cycles) and retention (10 years) have been experimentally demonstrated in the state-of-the-art. [R3.1–R3.4]. In this research, we have investigated the retention characteristics of the proposed multi-level cell and its impact on inference accuracy. Fig. R3.2 illustrates the measured V_{th} drift over time at two different temperatures for upto 24 hours. The randomly distributed states retention measurements (without verify) are carried out to understand the evolution of the V_{th} over time. The higher V_{th} states drift towards lower values whereas the lower V_{th} states remain fairly constant. Fig. R3.2(b) shows the target V_{th} retention measurements of the four chosen states at 85 °C.

To further investigate the consequence of such drift in the threshold voltage, we have now performed additional experimental measurements to analyze the DNN accuracy. The overlap in the distribution is calculated from the measured data which translates to a overlap in the MAC output. The new analysis shows that drifts due to FeFET retention results in merely 0.8% accuracy loss in the case of VGG19.

Next, we present the results obtained from endurance measurements. Two sets of endurance measurements are conducted. In the first case, we performed standard program/erase using voltage

Figure R3.2: Measurements for retention characteristics of the proposed 2-bit per cell FeFET. (a) Retention at 25 °C for random V_{th} states without target programming (b) Retention at 85 °C for target V_{th} states.

Figure R3.3: Distribution of the threshold voltage states at $t=0$ s and at $t=10^5$ s showing minimal overlap between the states.

Figure R3.4: Write endurance measurements of the proposed FeFET for (a) standard Program/Erase using fixed voltage pulses, and (b) target V_{th} programming with a write-verify scheme.

pulses of 4.5V (for programming) and -5V (for erasing) with a pulse duration of 50 ns. This process demonstrates an endurance of over 10^4 cycles. In the subsequent set of measurements, we assessed the endurance under target program conditions. This setup resembles multi-level-cell endurance since we utilize the same writing scheme, involving standard erase followed by target programming with increasing pulse amplitude and verifying V_{th} below the desired level or until a fixed number of pulses has been reached. The results also exhibit an endurance of at least 10^4 cycles. The reported endurance is relatively lower owing to the write-verify scheme employed. Once we do not reach the target anymore the algorithm always runs till the maximum allowed write pulse count and voltage. This results in accelerated degradation. Proper selection of the targets could help to increase endurance (while having the tradeoff of a smaller usable memory window). Also, a better write algorithm that maybe detects that the target value won't be reached anymore after a couple of tries and therefore limiting the write voltage from there on could improve endurance

However, since our focus is solely on neural network inference applications where memory states are written only once to store the neural network weights, a lower write endurance is still acceptable and results in no loss of inference accuracy as long as the data is retained.

Changes in the revised manuscript: In the revised manuscript, we have added the above discussions and experimentally measured retention and endurance characteristics. We have also reported the impact of the V_{th} drift on the inference accuracy. The new text is added from lines 300 to 310 and 345 to 349. Figure 6 is added in the main text. Figure S6 is added to the supplementary information.

R3Q3 - Additional clarification is needed to explain the reduction in power consumption resulting from including 1R, as mentioned in lines 118 to 120 on page 8.

R3A3 - As the power consumption is directly proportional to the current, the usage of our current limitation approach allows for lower accumulated current and, consequently, overall lower power consumption.

Changes in the revised manuscript: We have added the above clarification in the main text from lines 121 to 124.

R3Q4 - In lines 228 and 231 of page 16 of the manuscript, it is observed that the linearity of the V_{sampling} value for MAC output diminishes as the number of cells increases. The authors attribute this phenomenon to Equation 2; however, additional explanation is warranted to further clarify the application limitations of the proposed IMC crossbar macro. Please ensure a more detailed explanation of this observation, addressing the reasons behind the loss of linearity in the V_{sampling} value with an increasing number of cells.

R3A4 - We thank the reviewer for the valuable input. When considering larger arrays, the sampling voltage starts to saturate because as the number of cells increases, the current flowing through the source line increases for higher multiply-accumulate (MAC) output values. Consequently, the capacitor charges to its maximum value following an exponential charging relationship. Additionally, increasing the number of cells reduces the total resistance (since each cell is in parallel), resulting in

a decrease in the time it takes to reach the maximum output, leading to saturation.

This, in turn, poses a trade-off between the maximum operational speed and the capacitance value. Consequently, the capacitance value is carefully selected to prevent the sampling voltage from saturating while achieving the highest possible operational speed. It is worth noting that the realization of large capacitances often requires a significant amount of area, thus introducing another trade-off between speed and design area.

Changes in the revised manuscript: In the revised manuscript, we have added the above explanation from lines 237 to 249 in the main text.

References

- [R3.1] A. J. Tan, Y.-H. Liao, L.-C. Wang, N. Shanker, J.-H. Bae, et al., “Ferroelectric hfo₂ memory transistors with high- interfacial layer and write endurance exceeding 10¹⁰ cycles,” *IEEE Electron Device Letters*, vol. 42, no. 7, pp. 994–997, 2021.
- [R3.2] H. Mulaosmanovic, E. T. Breyer, S. Dünkel, S. Beyer, T. Mikolajick, et al., “Ferroelectric field-effect transistors based on hfo₂: A review,” *Nanotechnology*, vol. 32, no. 50, p. 502 002, 2021.
- [R3.3] B. Zeng, M. Liao, Q. Peng, W. Xiao, J. Liao, et al., “2-bit/cell operation of hf_{0.5}zr_{0.5}o₂ based fefet memory devices for nand applications,” *IEEE Journal of the Electron Devices Society*, vol. 7, pp. 551–556, 2019.

[R3.4] F. Müller, S. De, R. Olivo, M. Lederer, A. Altawil, et al., “Multilevel operation of ferroelectric fet memory arrays considering current percolation paths impacting switching behavior,” *IEEE Electron Device Letters*, vol. 44, no. 5, pp. 757–760, 2023.

REVIEWER COMMENTS

Reviewer #2 (Remarks to the Author):

Authors have addressed my concern, and I am ok with this revised manuscript.

Reviewer #3 (Remarks to the Author):

In the manuscript, the authors conducted simulations of a highly accurate IMC crossbar macro by minimizing variability effects between cells through the utilization of the 1FeFET-1R cell structure. In order for this manuscript to be considered for publication in Nature Communications, minor revisions are required. The specific details for these revisions are outlined below:

1. Additional discussion concerning the reduction of power consumption on lines 121 to 124 of page 8 appears necessary. When integrating a high resistance (1R) into a circuit, there is a general expectation that power consumption would increase to achieve comparable operation. If there is a comparison reference where power consumption is reduced more than in any other structure, highlighting this would further clarify the advantages asserted by the authors regarding the 1FeFET-1R concept.
2. A detailed explanation of the specific method employed for measuring the random V_{th} state and the target V_{th} state during retention measurements in Figure R3.2 is necessary. Provide sufficient explanation regarding how these values were derived, elucidating the procedures undertaken.

Letter and List of Updates

First Demonstration of In-Memory Computing Crossbar using Multi-level Cell FeFET

Nature Communications, <NCOMMS-23-21473A>

– Revision –

In the following, we provide a point-by-point response to the remaining two questions from reviewer 3. Changes in the manuscript are explicitly mentioned here and highlighted in the revised manuscript in blue. Reference, figure, and other numbers associated with this response letter are preceded by the letter “R”.

Summary of changes:

1. We have added further clarifications on the reduction of the cell power and performed a new analysis to demonstrate the effects that the included resistor has on limiting the cell current.
2. We have provided a detailed explanation clarifying the specific method that we employed for measuring the V_{th} states in the retention measurements as well as a detailed explanation for the involved procedures.

Reviewer 3

In the manuscript, the authors conducted simulations of a highly accurate IMC crossbar macro by minimizing variability effects between cells through the utilization of the 1FeFET-1R cell structure. In order for this manuscript to be considered for publication in Nature Communications, minor revisions are required. The specific details for these revisions are outlined below:

We sincerely thank the reviewer for the effort. In the following, we provide a point-by-point response to two remaining questions.

R3Q1 - Additional discussion concerning the reduction of power consumption on lines 121 to 124 of page 8 appears necessary. When integrating a high resistance (1R) into a circuit, there is a general expectation that power consumption would increase to achieve comparable operation. If there is a comparison reference where power consumption is reduced more than in any other structure, highlighting this would further clarify the advantages asserted by the authors regarding the 1FeFET-1R concept.

R3A1 - We agree with the reviewer that including a high resistance will increase the power ($P = I^2 \times R$), but only if the current flowing through the circuit remains constant. However, when the voltage is constant, then integrating a high resistance reduces the current. In our design, the 1FeFET-1R cell is connected to a constant supply voltage V_{dd} , which leads to a smaller current. Therefore, the power consumed by the cell is reduced when a high resistance is incorporated ($P = V^2/R$). In practice, when we connect a resistance to the FeFET device to form our 1FeFET-1R cell, this limits

the ON current to only $0.1 \mu\text{A}$, instead of $2 \mu\text{A}$, which is ON current in the baseline case where no resistance is connected to the FeFET. This is illustrated in Fig. R3.1, which presents the $I_D - V_G$ transfer characteristic of a particular stored state of the FeFET when the FeFET is connected to a resistance (dashed line) and not connected to a resistance (solid line). This observation is also aligned with existing works (e.g., [R3.1, R3.2]), which previously demonstrated that including a resistance leads to a lower current. Finally, it is noteworthy that the total power consumption of the entire crossbar array circuit is dominated by the power consumption of the analog-to-digital converters (ADCs). Therefore, reductions in the power of individual cells often do not lead to a noticeable impact on the total power consumption [R3.3, R3.4].

Figure R3.1: The effect of including a resistor on reducing the cell current from over $2 \mu\text{A}$ to only $0.1 \mu\text{A}$. This, in turn, leads to a smaller power consumption.

Changes in the revised manuscript: We have explained the reduction in power in page 8 at lines 122 to 124. In addition, we included the detailed discussion on the impact of resistance on power in the supplementary Section S1.

R3Q2 - A detailed explanation of the specific method employed for measuring the random V_{th} state and the target V_{th} state during retention measurements in Figure R3.2 is necessary. Provide sufficient explanation regarding how these values were derived, elucidating the procedures undertaken.

R3A2 - We apologize for the missing details and explanation regarding the employed method for measuring the different V_{th} states. In the following, we elucidate the employed method step by step as well as the different procedures employed for measuring the V_{th} states, programming and verifying the different V_{th} states as well as measuring the retention of FeFET devices over time.

① **Structure of the fabricated FeFET-based crossbar array:** The measurements are conducted on a fabricated FeFET-based crossbar array with a size of (9×7) . Hence, the array consists of 9 word-lines (WL) and 7 bit-/source-lines (BL/SL). To form a crossbar, the FeFET devices are “AND” connected, as Fig. R3.2 illustrates. All measurements are performed at the wafer level and the temperature desired for experiments is set using a temperature-controlled chuck.

② **Procedure of V_{th} measurement:** After a write voltage pulse is applied to an FeFET, the device exhibits a certain V_{th} . Reading-out the programmed/stored V_{th} state is necessary to 1) verify after writing what the exact V_{th} state, stored in FeFET, is and 2) perform later the required retention measurements, which quantify how the stored V_{th} states may drift over time. To extract the V_{th} of FeFET, the $I_D - V_G$ transfer characteristic of FeFET needs to be first measured. To perform that for FeFETs in a certain row, the corresponding WL voltage for that row is ramped from 0 V to 1.4 V with an increment of 100 mV, while applying 100 mV at the BL. The drain current I_D is

Figure R3.2: Structure of the AND-connected FeFET-based crossbar array (9×7) with 9 word-line (WL), 7 bit-line (BL) and 7 source-line (SL).

then sampled at every read voltage step V_G , with a sampling time of $80 \mu\text{s}$, until obtaining the full $I_D - V_G$ characteristic. To ensure a reliable current measurement, a settling time of $1 \mu\text{s}$ is waited for every read voltage step. Finally, the V_{th} state is extracted from the measured $I_D - V_G$ curve using the standard constant-current method [R3.5] in which the gate voltage is extracted at a certain fixed I_D current of 100 nA .

③ **Programming FeFET procedure:** Writing FeFET devices occur at the row level in which a specific write voltage is applied to a certain WL, while all SL/BL are set to 0 V . To ensure programming FeFETs to a certain targeted V_{th} state, a “write-verify” scheme is applied as follows. First a write voltage of 2.1 V is applied for 400 ns (i.e., a write pulse of 2.1 V amplitude and 400 ns pulse width is applied). Then, a period of 2 s is waited to provide a sufficient time for

Figure R3.3: Overview flowchart of the employed procedure to program FeFETs into a certain target V_{th} state along with the “write-verify” scheme.

any de-trapping within the FeFET. Afterwards, the V_{th} is measured to verify whether it matches the targeted level. V_{th} measurement is performed using the procedure explained in (2). If the measured V_{th} does not match the target level, then a new write voltage pulse, with 40 mV higher amplitude, is applied. The “write-verify” scheme is repeated until the target V_{th} is reached. It is important to note that once a specific FeFET device reaches the target V_{th} , it is changed to the “inhibit-condition” to ensure no disturbance occurs when other FeFET devices are being programmed. Inhibit is defined as a $V_{BL} = V_{SL} = 3.2$ V. To avoid disturbs in FeFETs sharing inhibited BL/SL along the passive WL, they are raised to $V_{WLp} = 1.6$ V. “Write-verify” scheme is continued until all 7 FeFETs of the activated/selected WL reach the target V_{th} state. The previous programming procedure is then applied to the another row (i.e., WL) to program the FeFET devices there to another target V_{th} state.

④ **Retention measurement procedure:** The focus of this work is to demonstrate how 2-bit FeFET can be employed to perform in-memory computing crossbar. To realize 2-bit FeFET, four different V_{th} states should be reliably stored. Therefore, we perform retention measurements to quantify how stored V_{th} states may drift over time. To this end, four different V_{th} states (1.2 V, 0.9 V, 0.6 V and 0.3 V) are targeted to be programmed in the FeFET devices. First, the temperature desired for the experiment (85 °C) is set using the temperature-controlled chuck and then a sufficient time is waited to ensure thermal stability. Then, four rows in the crossbar array are selected and the 7 FeFET devices in each row are programmed to a certain V_{th} state (1.2 V, 0.9 V, 0.6 V, 0.3 V). The FeFET programming procedure along with the “write-verify” scheme is explained in ③. Afterwards, the V_{th} states stored in the 7 FeFETs across one row are, in parallel, measured.

Figure R3.4: Retention measurements for the four stored V_{th} states. The “write-verify” scheme is applied during the FeFET programming to ensure the stored V_{th} state matches the target V_{th} state.

The V_{th} measurement procedure is explained in (2). The V_{th} measurements are then repeated with logarithmic time steps for 10^5 seconds, which is approximately one day. The obtained measurements provide the necessary information about the retention behaviour of FeFET devices (i.e., the drift of V_{th} over time). Fig. R3.4 presents the retention measurements of the four targeted V_{th} states.

It is noteworthy that without applying a “write-verify” scheme, the programmed V_{th} states will tend to be random due to the effects of variability. Fig. R3.5 demonstrates the retention measurements for 9 different V_{th} states that were programmed without applying the “write-verify” scheme. In this experiment, 9 different write voltages are applied to the 9 rows (i.e., WLs) in the crossbar array. The write voltages are selected to cover the entire switching range of FeFETs and they start from 2.3 V up to 3.1 V with an incremental step of 100 mV.

Figure R3.5: Retention measurement of for different 9 V_{th} states, which are random because the no “write-verify” scheme was applied during the FeFET programming, unlike the analysis in Fig. R3.4.

In this experiment, we employ the same V_{th} measurement procedure explained in ② and the same programming procedure explained in ③ but *without* the “write-verify” scheme.

Changes in the revised manuscript: We have included the above detailed explanation and procedures to the supplementary Section S3.

References

- [R3.1] D. Saito, T. Kobayashi, H. Koga, N. Ronchi, K. Banerjee, et al., “Analog in-memory computing in fefet-based 1t1r array for edge ai applications,” in *2021 Symposium on VLSI Technology*, 2021, pp. 1–2.
- [R3.2] A. Kazemi, F. Müller, M. M. Sharifi, H. Errahmouni, G. Gerlach, et al., “Achieving software-equivalent accuracy for hyperdimensional computing with ferroelectric-based in-memory computing,” *Scientific reports*, vol. 12, no. 1, p. 19 201, 2022.
- [R3.3] T. Soliman, N. Laleni, T. Kirchner, F. Müller, A. Shrivastava, et al., “Felix: A ferroelectric fet based low power mixed-signal in-memory architecture for dnn acceleration,” *ACM Transactions on Embedded Computing Systems*, vol. 21, no. 6, pp. 1–25, 2022.
- [R3.4] S. De, F. Müller, N. Laleni, M. Lederer, Y. Raffel, et al., “Demonstration of multiply-accumulate operation with 28 nm fefet crossbar array,” *IEEE Electron Device Letters*, vol. 43, no. 12, pp. 2081–2084, 2022.
- [R3.5] Y. Tsididis, *Operation and Modeling of the MOS Transistor*. McGraw-Hill, Inc., 1987.

REVIEWERS' COMMENTS

Reviewer #3 (Remarks to the Author):

The authors utilized the 1FeFET-1R cell structure of the manuscript to design a high-efficiency MAC circuit macro. They appropriately revised the explanation of the principle for reducing power consumption in the memory unit cell and the retention measurement method based on the reviewer's feedback. It seems that additional citations are needed for the manuscript explaining the fundamental principles of FeFETs and their memory and synaptic functionalities. Therefore, please include the following paper, which provides a detailed explanation of the principles of FeFETs and elucidates the memory and synaptic functionalities of FeFETs fabricated based on the ferroelectric characteristics.

1. Kim, J. Y. et al. Ferroelectric field effect transistors: Progress and perspective. *APL Mater* 9, 021102 (2021).

First Demonstration of In-Memory Computing Crossbar using Multi-level Cell FeFET

Nature Communications, <NCOMMS-23-21473B>

– Final Revision –

Reviewer #3 (Remarks to the Author):

The authors utilized the 1FeFET-1R cell structure of the manuscript to design a high-efficiency MAC circuit macro. They appropriately revised the explanation of the principle for reducing power consumption in the memory unit cell and the retention measurement method based on the reviewer's feedback. It seems that additional citations are needed for the manuscript explaining the fundamental principles of FeFETs and their memory and synaptic functionalities. Therefore, please include the following paper, which provides a detailed explanation of the principles of FeFETs and elucidates the memory and synaptic functionalities of FeFETs fabricated based on the ferroelectric characteristics.

1. Kim, J. Y. et al. Ferroelectric field effect transistors: Progress and perspective. *APL Mater* 9, 021102 (2021).

We would like to thank the reviewer for the valuable feedback, which we received over the past revisions. As proposed by the reviewer, we have now revised the final manuscript and included the suggested reference.

The new citation “*Kim, J. Y., Choi, M.-J. & Jang, H.W. Ferroelectric field effect transistors: Progress and Perspective. APL Materials 9 (2021)*” has the index number 9 and it is referred in the manuscript in page 5.